# A Novel Neighbor Housing Environment Enhances Social Interaction and Rescues Cognitive Deficits from Social Isolation in Adolescence

**DOI:** 10.3390/brainsci9120336

**Published:** 2019-11-22

**Authors:** Alexander B. Pais, Anthony C. Pais, Gabriel Elmisurati, So Hyun Park, Michael F. Miles, Jennifer T. Wolstenholme

**Affiliations:** 1VCU-Alcohol Research Center, Virginia Commonwealth University, Richmond, VA 23298-0613, USA; paisab@vcu.edu (A.B.P.); paisac@vcu.edu (A.C.P.); elmisuratiga@vcu.edu (G.E.); michael.miles@vcuhealth.org (M.F.M.); 2Department of Pharmacology and Toxicology, Virginia Commonwealth University, Richmond, VA 23298-0613, USA; julie.park@rice.edu

**Keywords:** social isolation, adolescence, recognition memory, anxiety-like behavior, social interaction

## Abstract

Adolescence is characterized by high levels of playful social interaction, cognitive development, and increased risk-taking behavior. Juvenile exposure to social isolation or social stress can reduce myelin content in the frontal cortex, alter neuronal excitability, and disrupt hypothalamic pituitary adrenal (HPA) axis function. As compared to group housed animals, social isolation increases anxiety-like phenotypes and reduces social and cognitive performance in adulthood. We designed a neighbor housing environment to alleviate issues related to social isolation that still allowed individual homecages. Neighbor housing consists of four standard mouse cages fused together with semi-permeable ports that allow visual, olfactory, and limited social contact between mice. Adolescent C57BL/6J males and females were group housed (4/cage), single housed (1/cage), or neighbor housed (4/complex). As adults, mice were tested for social, anxiety-like, and cognitive behaviors. Living in this neighbor environment reduced anxiety-like behavior in the social interaction task and in the light-dark task. It also rescued cognitive deficits from single housing in the novel object recognition task. These data suggest that neighbor housing may partially ameliorate the social anxiety and cognitive deficits induced by social isolation. These neighbor cage environments may serve as a conduit by which researchers can house mice in individual cages while still enabling limited social interactions to better model typical adolescent development.

## 1. Introduction

Adolescence is a significant period of brain and behavioral development where peer to peer social interactions occur more frequently than during any other developmental period. The outcomes of this period on social, cognitive, and anxiety-like behavior are not unique to humans and occur in other species where adolescence is characterized by high levels of playful social interactions, increased risk-taking behaviors, and cognitive development [1]. In adolescent rats and mice, there is a marked increase of social interactions characterized by play behavior that serves as a learning experience and helps shape adult behaviors [2]. These peer–peer interactions peak during early adolescence and decline to low levels after sexual maturation [3]. Both beneficial and adverse social experiences during adolescence in male mice influence anxiety-like behaviors [4,5,6,7,8,9,10], depressive-like behavior [8,11], and cognition later in adulthood [5,8,12,13]. A wealth of research shows that experiences during the adolescence period are critical in modulating these behaviors and ultimately proper neural development and function [14,15]. For example, repeated social defeat induced social avoidance in both adolescent and adult mice, but only mice that were socially defeated during early adolescence displayed enduring problems with cognitive flexibility [13]. Chronic social isolation and/or chronic variable stress in adolescence, but not adulthood, increased ethanol drinking in adulthood [16]. Thus, the role of experience is critical to the developing neurocircuitry of the adolescent brain and can have lasting affects into adulthood in mice.

Investigations of the structural and molecular changes in the brain associated with adolescent social isolation have identified reduced neuronal excitability in the dentate gyrus [17,18], and white matter structural changes in the prefrontal cortex [19,20]. Mice that were socially isolated two weeks after weaning displayed cognitive and behavioral dysfunction that was associated with alterations in the white matter tracts of the prefrontal cortex [19]. Adolescent social isolation also lead to more immature dendritic spines, impaired plasticity in the prefrontal cortex [21], and altered the morphology of inhibitory circuits in the prefrontal cortex and basolateral amygdala [22]. Similar effects of social isolation can also be induced using much longer periods of single housing in adulthood, with adult mice showing that prolonged social isolation-induced behavioral, transcriptional and ultrastructural changes in oligodendrocytes in the prefrontal cortex, resulting in impaired myelination [23]. The prefrontal cortex is not the only brain region affected by adolescent social isolation. Singly housed adolescent mice displayed widespread changes across the brain with an altered structural connectome [24]. Social isolation has also been shown to disrupt the hypothalamus-pituitary axis function and increase anxiety-like behaviors in single housed mice as compared to their group housed counterparts [25]. A lack of social enrichment during the critical period of adolescence likely removes necessary stimulation from these brain regions, giving rise to impaired development of neural connections and structure, which persists into adulthood.

A common experimental design in rodent model studies is to individually house the subject with no interaction to a conspecific counterpart. Unless specifically testing the effects of social isolation stress, single housing is frequently necessary following surgical procedures, to conduct drug self-administration studies, to track other individual behaviors, or to eliminate risk of physical harm to the mouse due to territorial aggression, particularly in males [26]. The alternative is to group house the subjects in pairs or greater. These housing conditions, particularly single housing, create an artificial environment that is unnatural, socially and developmentally isolating, and can induce behavioral changes in mice, with stronger effects in adolescents than adults. Indeed, single housing is frequently used as an ethologically-relevant social stress that can be used to model post-traumatic stress [10,27], depression, and anxiety [8,11]. Conversely, in mice that are group-housed, experimental results are potentially confounded by factors such as social hierarchy or territorial aggression. The dichotomy of these two housing environments, and the behavioral and physiological changes they induce, poses problems with conducting research during the adolescence period and consequently the external validity of the findings.

These complications of housing are particularly relevant for alcohol or other drug abuse research that seeks to characterize the effect of a voluntarily consumed substance on behavior and the maturation of neural development or function. The results of a study with individually housed (i.e., socially stressed) animal models may represent an admixture of the environment effects and the substance administered. Unfortunately, most of these studies require single housing of animals in order to be able to record voluntary individual consumption and/or preference levels. Our current study, therefore, behaviorally characterizes a novel housing environment, neighbor housing, and the effect of three different housing conditions (single housing,1/cage; group housing, 4/cage; and neighbor housing, 4/complex) on the social, cognitive, and anxiety-like behaviors of C57BL/6J male and female mice exposed to these environments from adolescence into adulthood. We seek to answer whether neighbor housing, that allows for limited social and olfactory interaction with limited physical contact, can serve as an intermediate housing environment between the extremes of single housing and group housing and better model adolescent development when single housing is still necessary for experimental design.

## 2. Materials and Methods

### 2.1. Animals and Housing

Male and female C57BL/6J from Jackson Laboratory (Bar Harbor, ME, USA) arrived at post-natal day (PND) 22 and were housed in same-sexed cages (4/cage) for 1 week in an American Association for Laboratory Animal Care (AALAC)-accredited facility under 12-h light/dark cycles. On PND 29, the mice were housed in same-sex groups in one of three conditions (*n* = 12/group/sex) until the end of the experiment: group housed (GH; 4/cage), neighbor housed (NH; 4/complex) or single housed (SH; 1/cage). Our custom-made neighbor housed cages were constructed from 4 standard polycarbonate ventilated cages (28.5 × 17.5 × 12.5 cm) with 4 circular cut-out ports (6 cm in diameter) along the sides conjoined by 4 polycarbonate portals (Figure 1). Each portal was blocked with an 8 mm welded metal mesh with 1 cm^2^ wide openings that enabled limited physical interaction but full visual, olfactory and auditory stimuli from mice in the two conjoining cages. Thus, each neighbor-housed mouse had two same-sexed neighbors that remained constant. Each cage across housing condition was filled with woodchip bedding (Sani-Chips 7090, Teklad/Envigo, Cambridgeshire, UK) and 1 square cotton nestlet (2.5 g) was given at each cage change. Food (Teklad Diet 7012) and water was present ad libitum. Neighbor housed and single housed mice resided in the same room as their group housed counterparts and were exposed to similar olfactory, auditory, and visual stimuli. Experimenter interaction during this period of housing manipulation (from PND 29–58) was identical between groups and was restricted to bodyweight assessments every 2–3 days and weekly cage changes. Behavioral assay testing began on PND 58–59. All animal housing and care was conducted with the approval of the Virginia Commonwealth University IACUC Committee and in accordance with the NIH Guide for the Care and Use of Laboratory Animals [28].

### 2.2. Social Interaction Test

On PND 58–59, four weeks after housing assignments, mice were habituated to the test room for 1 h and then tested for social interaction under low light conditions during the dark cycle. This task is a modified version of the social interaction task as previously described [29]. Male and female mice (*n* = 12/group/sex) were habituated to an open field locomotor activity box (41 × 41 × 31 cm, Omnitech Electronics, Inc., Columbus, OH, USA) containing an empty fine metal mesh cylinder for 3 min with no stimulus mouse present. The open field box was topographically divided into an interaction zone (25 × 7.5 cm) surrounding the stimulus mouse, and two corner zones (10 × 10 cm) far away from the interaction zone. Tracking software (Fusion v5.3; Omnitech Electronics Inc., Columbus, OH, USA) was used to record movement and animal position via infrared photobeam breaks. Between the habituation and test, mice were returned to their home cage for 30-s, while an unfamiliar adult female C57BL/6J mouse was placed under the inverted fine metal mesh cylinder (8.5 cm diameter, 11 cm height). During the test phase, the amount of time and distance the test mouse traveled in the interaction zone or in the opposing corner zones was recorded for three minutes.

### 2.3. Novel Object Recognition

We used the novel object recognition task to measure prefrontal cortex (PFC)-mediated recognition memory, as previously described [30]. Novel object recognition involved a training and a test phase, separated by a 5-min delay. On PND 63–65, mice (*n* = 11–12/group) were habituated to the test cage for 30 min one day prior to the task. On test day, mice were habituated to the testing room for 1 h, then to the test cage for 30 min. During the training phase, mice were allowed to interact with two identical objects placed in opposite corners of an empty clean mouse cage, for 5-min. Mice were then returned to their home cage for a 5-min inter-trial delay to measure PFC-dependent short term memory [31]. During the inter-trial delay, one familiar object was replaced by a novel object of similar size. Mice were returned to the test arena and allowed to explore both objects for 5 min. Time in close contact with nose oriented towards the object (<2 cm) was recorded. Scorers were blinded to the sex and treatment of the mice. A discrimination index was calculated by subtracting the time interacting with the familiar object from the time interacting with the novel object divided by the total interaction time. Failure to spend more time with the novel object was interpreted as impaired recognition memory and PFC dysfunction [32]. Any mouse that did not investigate the objects for more than 10 s during training was not used in the analysis. No mice were excluded for this reason.

### 2.4. Anxiety-Like Behavior in the Light-Dark Box

At PND 77–78, mice (*n* = 11–12/group) were tested for differences in basal anxiety-like behavior during the dark phase of the light-dark cycle. The light-dark (LD) box conflict model for anxiety-like behavior was conducted using a standard commercial open field activity box divided into two equally sized light and dark zones (25.4 × 12.7 × 20.3 cm). Tracking software (Fusion v5.3; Omnitech Electronics Inc.) was used to record movement, as described [30]. Animal position and locomotor activity was monitored by infrared photobeam breaks. Following a 1-h acclimation period to the behavioral room, mice were placed in the center of the light chamber facing the entrance to the dark chamber. Studies consisted of a 5-min test session, initiated once the animal entered the dark compartment. Anxiety-like measures were reported as percent time spent in the light and percent distance traveled in light to control for locomotor activity. An increase in either measure was interpreted as decreased anxiety-like behavior. We also recorded the latency to enter the light. Increased latency to enter the light was interpreted as an increased anxiety phenotype.

### 2.5. Statistics

Bodyweight data were analyzed using a repeated measures two-way ANOVA with time and housing condition as factors. Behavioral data were measured using two-way ANOVAs with sex and housing conditions as factors. Student–Newman–Keuls post hoc tests were used to calculate significance. *p*-values less than 0.05 were considered significant.

## 3. Results

### 3.1. Social Housing Does not Alter Bodyweight

Male and female bodyweights were collected every 2–3 days during the period of housing manipulation (Figure 2). As expected, a main effect of sex was noted with males in all housing environments having greater bodyweights than females (F_5,1856_ = 23.351, *p* < 0.001). Thus, separate two-way repeated measures ANOVAs were run for each sex to determine an effect of housing on bodyweight. In both males (F_2,966_ = 0.740, *p* = 0.485) and females (F_2,889_ = 0.108, *p* = 0.898), housing did not alter bodyweight during the entire experiment. A main effect of time was noted in both males (F_26,966_ = 558.21, *p* < 0.001) and females (F_26,889_ = 498.089, *p* < 0.001), where bodyweight steadily increased over time.

### 3.2. Neighbor Housing Increases Social Interaction

A social interaction test was conducted to discern the social behavior of mice raised in separate housing conditions from adolescence to adulthood. Sociability is measured as an increased amount of time spent in the interaction zone interacting with a novel stimulus mouse as opposed to the corner zones (Figure 3). During the test phase, for time spent in the interaction zone, two-way ANOVA revealed a significant main effect of housing and sex, but no significant interactions were found (F_2,71_ = 0.418, *p* = 0.660). Neighbor housed mice spent significantly more time interacting with the stimulus mouse than group housed mice, but were not significantly different than single housed mice (F_2,71_ = 3.414, *p* = 0.039, Figure 3A). Not surprisingly, two-way ANOVA also revealed significant differences in sociability among sex, where male test subjects interacted more with the female stimulus mouse than female test subjects (F_1,71_ = 7.7753, *p* = 0.007).

The time spent in the corner zones during the test phase, was not significantly different between housing conditions (F_2,71_ = 1.228, *p* = 0.300, Figure 3B), but was significantly greater in females than males (F_1,71_ = 8.229, *p* = 0.006). No significant interactions were found (F_2,71_ = 0.320, *p* = 0.727). Importantly, the total locomotor activity in this task was not altered by housing condition (F_2,71_ = 2.131, *p* = 0.127) or sex (F_1,71_ = 2.537, *p* = 0.116, Figure 3C).

### 3.3. Neighbor Housing Rescues Social Isolation Deficits in Recognition Memory

A novel object recognition test was conducted to discern the cognitive ability of mice in separate housing conditions (Figure 4). A positive discrimination index indicates preference for the novel object, while a zero or negative index suggests impaired recognition memory and PFC dysfunction [31]. Two-way ANOVA revealed a main effect of housing (F_2,69_ = 20.696, *p* ≤ 0.001). Single housed mice did not discriminate between the novel and familiar objects, while group and neighbor housed mice displayed a positive discrimination index. There was no difference between group or neighbor housed mice in their ability to prefer the novel object. No significant main effect of sex (F_1,69_ = 0.261, *p* = 0.611), or interaction (F_2,69_ = 0.165, *p* = 0.848) was found. There were no significant differences between housing conditions (F_2,69_ = 0.089, *p* = 0.915) or sex (F_1,69_ = 0.438, *p* = 0.510) for time investigating the objects during the training session.

### 3.4. Anxiety-Like Behavior in the Light-Dark Box is Reduced by Neighbor Housing

Anxiety-like behavior was measured using a light-dark box apparatus to record the percent of time a mouse spent in the light area of the testing chamber (Figure 5). The more time spent in the lit arena as opposed to the dark arena suggests reduced levels of anxiety-like behavior in this behavioral paradigm. By two-way ANOVA, there was a significant increase in percent time in the light between the neighbor housed versus both single and group housed conditions (F_2,69_ = 5.786, *p* = 0.005). No significant differences were noted between sex (F_1,69_ = 0.0479, *p* = 0.827), nor the interaction of housing and sex (F_2,69_ = 0.152, *p* = 0.859). Two-way ANOVA also revealed a significant decrease in percent time spent in the dark (F_2,69_= 5.786, *p* = 0.005) in neighbor housed versus group or single housed mice. No significant differences were found for percent time in the dark for sex (F_1,69_ = 0.0479, *p* = 0.827), or for the interaction (F_2,69_ = 0.152, *p* = 0.859). The percent distance travelled in the light mirrored the percent time in the light (data not shown). A main effect of housing (F_2,69_ = 3.973, *p* = 0.024) was noted, but no significant effect of sex (F_1,69_ = 0.145, *p* = 0.704) or interaction (F_2,69_ = 0.181, *p* = 0.835) was found. Similarly, the percent distance travelled in the dark was also significantly affected by housing (data not shown). A main effect of housing (F_2,69_ = 3.973, *p* = 0.024) was found where neighbor housed mice travelled less in the dark than group or single housed mice. No effect of sex (F_1,69_ = 0.145, *p* = 0.704) or interactions (F_2,69_ = 0.181, *p* = 0.835) were found. Furthermore, single housed mice were more hesitant to enter the light-filled area of the box than group or neighborhood housed mice (F_2,69_ = 3.327, *p* = 0.042), as measured by latency to enter the lit arena. Latency to enter the light did not differ between sexes (F_1,69_ = 0.00173, *p* = 0.967), nor was there a significant interaction (F_2,69_ = 1.103, *p* = 0.338). On note, total locomotor distance traveled in the whole light-dark apparatus was significantly increased in neighbor housed mice as compared to group and single housed mice (F_2,69_ = 16.017, *p* < 0.001). No significant effects were found between sexes (F_1,69_ = 0.899, *p* = 0.347) or with an interaction between housing and sexes (F_2,69_ = 0.420, *p* = 0.0659).

## 4. Discussion

Here, we characterized the social, anxiety-like, and cognitive behavior of male and female mice in a novel neighbor housing complex and compared it to group or single housed mice. As compared to group housed or single housed mice, neighbor housing increased social interactions and decreased basal anxiety-like behavior. Neighbor housing or group housing did not alter recognition memory in the novel object task, but single housing lead to recognition deficits. Of note, sex did not interact with housing condition to alter these behaviors.

An advantage of the neighbor housing environment is that it allows for limited social and olfactory interaction, while giving each mouse its own living space. Due to the portal openings between two adjacent cages, visual stimuli, social odors and vocalizations from two different mice can enter the cage of an adjacent mouse. Each mouse can also have very limited physical interaction with the two other neighboring mice via the metal mesh partitions in the portals. Mice can interact through ultrasonic vocalizations and can recognize other mice through vision and urinary odors [3] which is essential for normal behavioral development [33]. Social, olfactory, and visual interactions are also present in the group housed condition but can be accompanied by territorial aggression and dominant/submissive phenotypes that can also alter affective behaviors [26]. These social cues are largely absent in single housed mice which creates a socially stressful situation, particularly when this occurs during adolescence [15].

Differences in bodyweight gain from single housing during adolescence are often reported [5,6,10] with socially isolated animals having slower weight gain during the individual housing period. But this is not always the case, as adolescent social isolation did not lead to any obvious differences in general health such as alterations in weight gain in male rats [9], in male or female C57BL/6J mice [7], or in male DBA/2J mice [6]. In the present study, we also found that C57BL/6J mice, regardless of housing environment, all gained weight at the same rate. Strain and/or species genetic differences may account for this discrepancy. Additionally, our studies included a small cotton nestlet in each mouse’s cage. This form of “environmental enrichment” was sufficient to potentiate the increased drinking of single housed C57BL/6J mice, since isolated mice housed in an enriched environment (i.e., containing a cotton nestlet) did not drink more ethanol than control mice that were group housed [7]. Bodyweight was also not different in any of these housing conditions with or without enrichment [7], suggesting that some aspects of single housing (i.e., bodyweight gain and ethanol drinking) can be mitigated by a simple form of environmental enrichment.

We used two assays to investigate the role of adolescent housing on anxiety-like behavior, the social interaction test and the light-dark box. The social recognition task was used to measure the social aspects of anxiety-like behavior such as social approach or social avoidance [29]. We observed a significant increase in time interacting with the stimulus mouse in neighbor housed mice as compared to group housed mice, suggesting a lower social anxiety phenotype. Few reports have directly investigated effects of social isolation on social and affiliative behaviors, and those that have offer conflicting results [5,34,35,36,37,38]. Using a similar social interaction task, C57/BL6 males socially isolated since weaning interacted less with an aggressive male conspecific as compared to group housed mice [35]. While we did not observe a difference in social interaction between single and group housed mice, this discrepancy may be explained by the difference in stimulus mice used in our assay since we used a non-aggressive female stimulus. Using the social recognition task to assess social memory in male rats [38] or male and female mice [36], social isolation from mid-adolescence into adulthood attenuated the expected habituation to repeated exposures to a stimulus female. Socially isolated animals also failed to recognize a novel stimulus suggesting impaired recognition memory following adolescent isolation. Using the social preference task for a novel mouse over an empty cylinder, Lander et al. [5] found that single housing during adolescence increased social investigation of a novel mouse as compared to group housed adolescents, while Medendorp et al. [21] showed no preference for the novel mouse over an empty cylinder. In adolescent rats, social isolation increased aggression but had no effect on social interaction, while adult social isolation increased social interactions with a conspecific [34]. In C57BL/6J mice, social isolation from weaning until adulthood increased social interactions with a novel mouse over a familiar mouse in the social preference task [37].

In the light-dark assay, neighbor housed mice also showed reduced anxiety-like behaviors. They spent more time and traveled a greater distance in the light-filled arena than single or group housed mice. Total locomotor distance travelled during this assay was also greater in neighbor housed mice. Increased total exploratory behavior in the light-dark assay may also involve other factors such as exploratory drive (curiosity) and emotional reactivity (i.e., fear or anxiety-like behavior) to a novel arena and interpretation of this as a proxy for general locomotor activity has been cautioned [39,40]. Increased activity in this task, however, can be confounding. Further investigation of the observed increased locomotor activity is needed in the neighbor housed mice. The latency to enter the light, however, was significantly greater in single housed mice, suggesting that social isolation increased some aspects of anxiety-like behavior in this task. Social isolation frequently produces inconsistent results on anxiety-like behavior and the general consensus is that positive results are dependent on the species and type of assay used. For example, social isolation of C57BL/6 males in either adolescence or adulthood increased anxiety-like behavior in the open field [5] and increased anxiety in the elevated plus maze in adolescent ICR mice [8] and in adolescent rats [9,10]. However, rearing in social isolation during adolescence decreased anxiety-like behavior in the elevated plus maze in C57BL/6J and DBA/2J mice [6] and in the light-dark test in C57BL/6J mice [7]. Thus, the effects of social isolation on anxiety-like behavior are conflicting. Here, we observed increased latency for singly housed mice to enter the light arena of the light-dark box. Neighbor housed mice, however, showed significant increases in both the time interacting with the stimulus mouse and time in the light, suggesting lower anxiety phenotypes than either single or group housed mice. Single housed mice and group housed mice did not differ on these measures in the light dark task.

This overall decrease in anxiety-like behaviors (i.e., increased social interaction and increased activity in the light-dark box) in neighbor housed mice could alternatively suggest that neighbor housing induces a general hyperactivity and/or increases novelty seeking [5]. However, these potential increase in activity or novelty-seeking were not consistently found in all of the behavioral assays. For example, locomotor activity did not differ between housing conditions in the social interaction task. The latency to enter the light-filled arena also did not differ between group and neighbor housed mice. Neighbor and group housed mice spent equal lengths of time exploring the newly-presented objects during the first session of the novel object recognition test, and also exploring the novel object in the second session. Together, these data suggest that a general increase in novelty seeking was not likely. A second alternative explanation is that greater locomotor activity during the light-dark test and investigation of an unfamiliar social stimulus reflects behavioral disinhibition, associated with enhanced striatal dopamine. Frequent stressful life events in childhood has been associated with hyperactivity in the HPA axis and may play a role in the modulation of the onset of anxiety in humans [41]. Likewise, in rodent models, single housing can disrupt HPA axis functioning [25], and increase the glutamatergic tone in the frontal cortex [5]. Neighbor housing may serve as an intermediary between group housing and social isolation that enables mice to have increased positive interactions (decreased bullying and other territorial aggressions) that enhance their resiliency and reduce the chance for disruptions in HPA axis and cortical development during adolescent development. Further investigations are needed to explore the brain regulatory mechanisms that may or may not be fully rescued in the neighbor housing condition.

Exposure to social stress or social isolation in adolescence is associated with alterations in glutamatergic and GABAergic transmission [5,17,18], myelin-related gene expression [19], structural connectivity and impaired plasticity [21]. These changes in the frontal cortex may be related to the memory deficits frequently observed in socially isolated animals [5,12,42]. Novel object recognition [5,6,8,42] and fear conditioning [10] show deficits after adolescent social isolation. Adolescent social isolation can also lead to low spatial learning and memory in the Morris water maze and poor working memory in the Y-maze [12]. In the present study, the ability to discern between a novel and familiar object was greatly impaired in single housed mice. Object recognition memory was not impaired in group housed mice and neighbor housing returned object recognition memory to the levels observed in group housed mice. What remains unclear is if these memory deficits are a result of problems with memory consolidation or with enhanced forgetting [10,42]. Future studies are needed to address this question.

An enriched physical environment in adolescent socially isolated mice rescues hippocampal-related spatial memory, but does not enhance the effects on the PFC-related abnormalities of social preference and memory [12]. The authors suggest a brain region-specific effectiveness of enriched physical environment in remediating brain impairment of isolated mice, with a complete reversal of hippocampal structural impairments and cognitive dysfunctions, but without mitigation of mPFC pathological changes or associated anxiety and social interaction defects [12]. Re-socialization with other group housed conspecifics after adolescent social isolation, however, can restore the impaired social interaction and hypomyelination in the medial prefrontal cortex [20], suggesting a social component is necessary for complete restoration of typical behavioral responses following social isolation. The goal of our neighbor housing is to minimize the adverse effects a single housing environment can induce on behavior and physiological changes in the brain during adolescence and to minimize the chance for harm that can occur in group housing conditions. Here, we show that neighbor housing offers a positive alternative to single housing that would enable mice to remain in their own housing environment. This novel housing reduced social and basal anxiety without altering novel object recognition memory as was observed with social isolation.

We believe that the novel neighborhood housing model serves to fulfill one of the three R’s of animal research: Replacement, Reduction, and Refinement [43], better than the group or single housing conditions. In regards to Refinement, the neighborhood housing increases animal welfare as evidenced by reduced anxiety-like behaviors and normalized cognitive behaviors when compared to singly housed mice. In addition, the neighborhood housing environment eliminates the chance of physical harm that can be induced on a mouse from its counterparts due to aggression, which is a common occurrence in the group house environment. It may also reduce the influence of dominant/submissive phenotypes, which could reduce variability in affective measures of behavior. Also, in alcohol and drug abuse research in which drug seeking and consumption is considered to have a social component [44,45,46], the neighbor housing environment allows mice to interact with two other conspecifics while enabling the ability to easily individually quantify the amount of drug consumed for a specific mouse without expensive monitoring equipment. The benefits conferred by this housing condition on drug abuse research could also extend into other pharmacological studies that wish to observe an animal in a conspecific group while minimizing any aversive behaviors such as bullying. We believe that the neighborhood housing environment can serve as an intermediary that tempers the effect of the environment on behaviors in adolescent mice that are present in single housing, and increases animal welfare and safety through eliminating the chance for injury or harm through aggression that is present in the group housed environment. Further studies are needed to characterize the brain-regional and molecular signaling effects of this novel housing environment.

## 5. Conclusions

Adolescence is a period of high peer-to-peer social interactions that are critical for proper brain and neurobehavioral development (Spear 2000). Social isolation during adolescence alters neuronal excitability and leads to lasting structural changes in the brain. We designed and characterized a novel neighbor housing environment to alleviate issues related to social isolation that still allowed individual home cages. Our neighbor housing complexes consist of four standard mouse cages fused together with semi-permeable ports that allow visual, olfactory, and limited physical contact between mice. Neighbor housing increased social interactions, decreased basal anxiety-like behavior, and did not alter recognition memory in the novel object task. This novel neighbor housing environment minimizes the adverse effects of single housing and eliminates the chance for harm that can occur in group housing conditions. Thus, it may serve as a solution in cases where single housing is needed but not desired.

## Figures and Tables

**Figure 1 brainsci-09-00336-f001:**
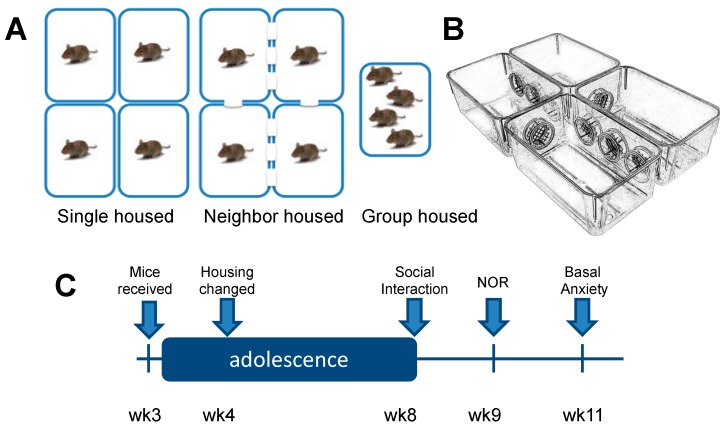
Experimental Design (**A**) Male (*n* = 36) and female (*n* = 36) C57BL/6J mice were housed in same-sex groups in one of three conditions from post-natal day 29 (PND 29) until the end of the experiment: Group housed (GH; 4/cage; *n* = 12), Neighbor housed (NH; 4/complex; *n* = 12) or Single housed (SH; 1/cage; *n* = 12). Food and water were available ad libitum. (**B**) The neighbor housing complex consists of four standard mouse cages fused together with polycarbonate ports divided by an 8 mm mesh allowing for minimal physical and olfactory contact. (**C**) Behavioral testing began on PND 58. The assays tested were social interaction, novel object recognition (NOR), and anxiety-like behavior in the light-dark box. At least four days elapsed between each behavior assay.

**Figure 2 brainsci-09-00336-f002:**
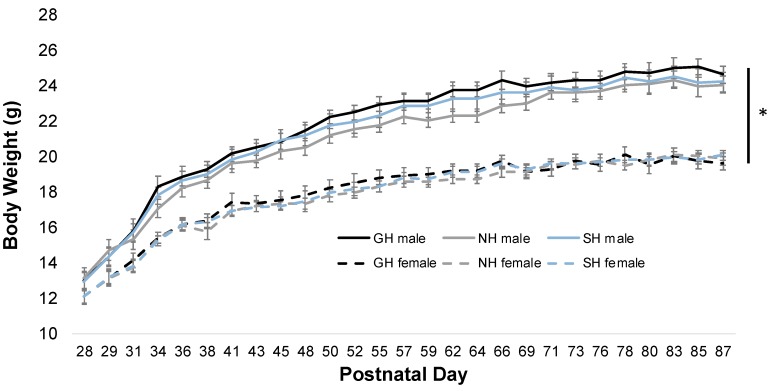
Housing condition does not alter bodyweight. While bodyweights significantly increased over time, housing condition did not alter this trajectory. Males weighed significantly more than females, * *p* < 0.001, main effect of sex by two-way RMANOVA.

**Figure 3 brainsci-09-00336-f003:**
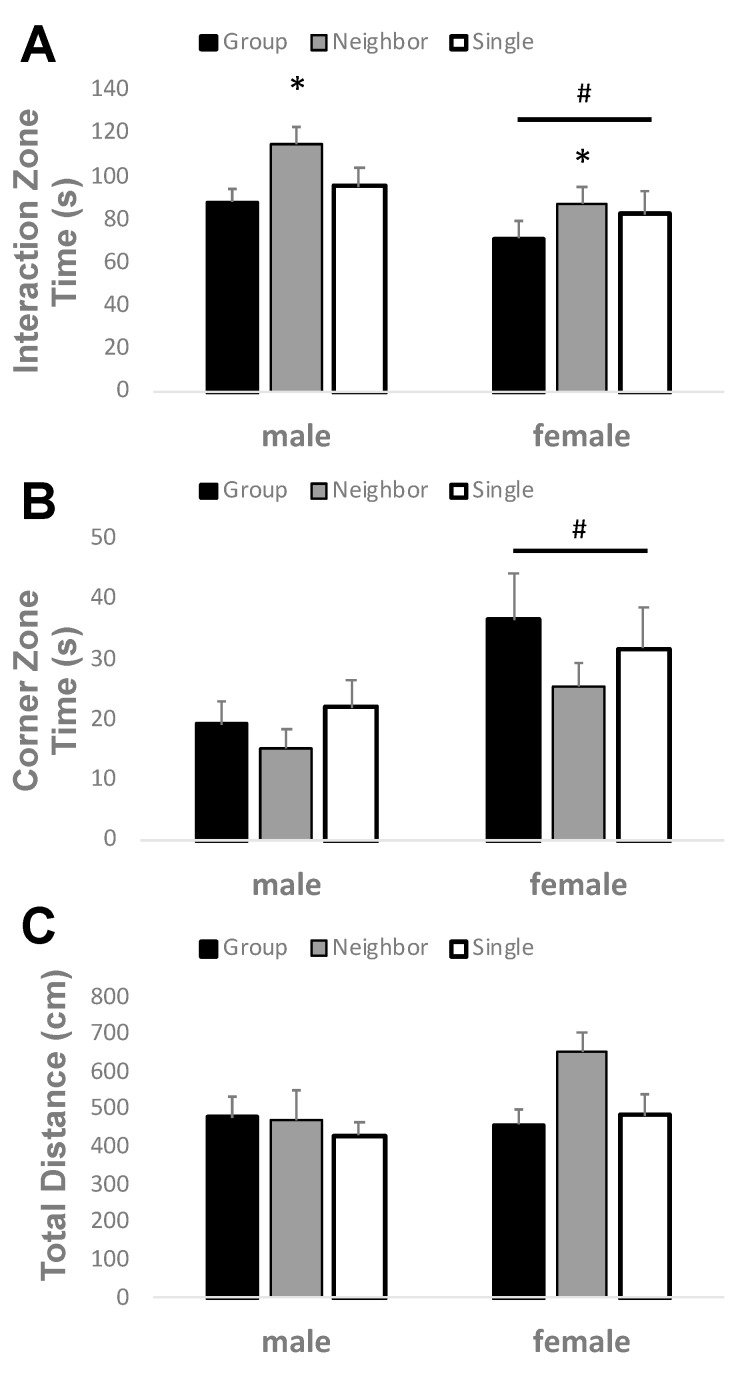
Neighbor housing increases social interaction in both males and females. (**A**) Time spent in the interaction zone with a stimulus mouse was significantly increased in neighbor housed mice as compared with group housed mice. Males spent more time in the interaction zone than females, regardless of housing condition. (**B**) Time spent in the corner zones far away from the stimulus mouse was not significantly altered by housing, but was increased in females as compared to males. (**C**) Total distance travelled in the arena was not significantly altered by housing condition or sex. * *p* < 0.05 by two-way ANOVA versus group housed; ^#^
*p* < 0.05 versus males by two-way ANOVA.

**Figure 4 brainsci-09-00336-f004:**
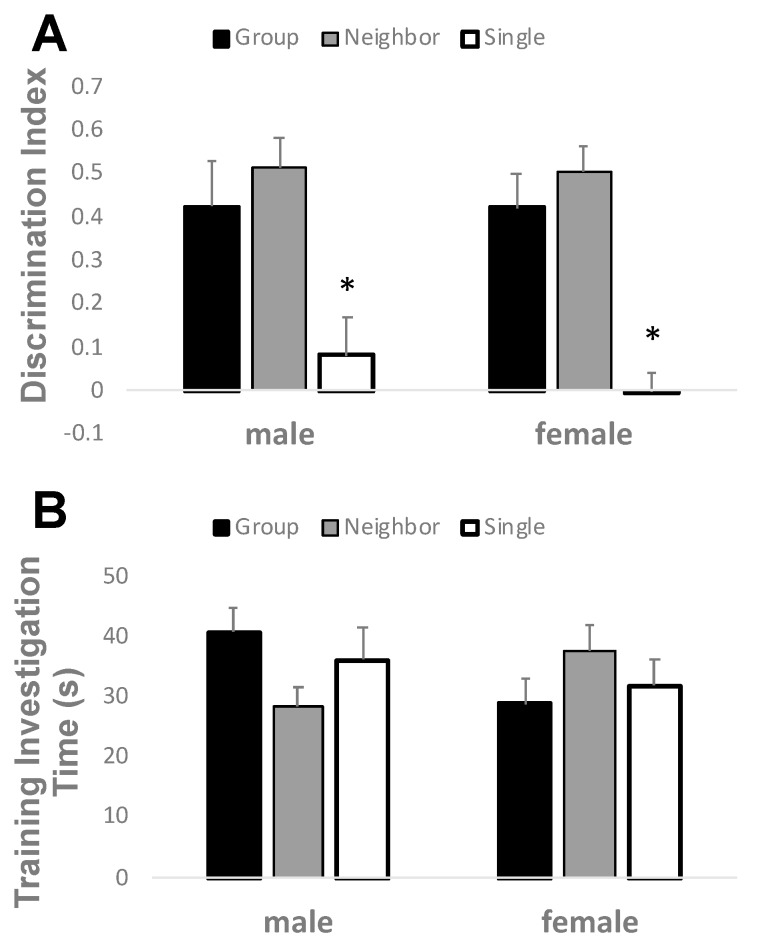
Neighbor housing rescued cognitive deficits in novel object recognition memory. (**A**) Using a 5 min inter-trial interval, single housed mice had a lower discrimination index as compared to group and neighbor housed mice. (**B**) Total investigation time during training did not differ between groups or sex. * *p* < 0.001 by two-way ANOVA for main effect of housing.

**Figure 5 brainsci-09-00336-f005:**
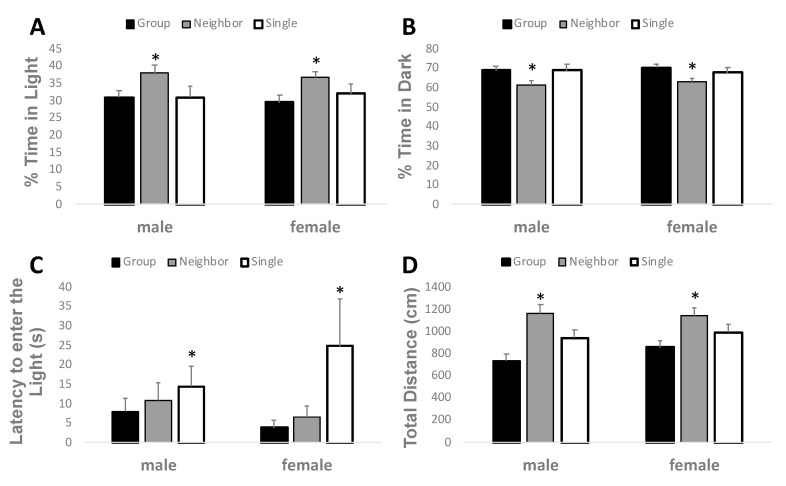
Neighbor housing reduces anxiety-like behavior in the light dark box. Percent time in the light (**A**), percent time in the dark (**B**), and total distance travelled (**C**) were significantly increased in neighbor housed mice as compared to group or single housed mice, (**D**) The latency to enter the light was significantly increased by single housing. * *p* < 0.05 versus the other two housing conditions by two-way ANOVA main effect of housing.

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
