# Peer review of "A Novel Neighbor Housing Environment Enhances Social Interaction and Rescues Cognitive Deficits from Social Isolation in Adolescence"

_brainsci, 2019, doi:10.3390/brainsci9120336_

Round 1

Reviewer 1 Report

The  paper “A novel neighbor housing environment enhances social interaction and rescues cognitive deficits from social isolation in adolescence ” written by Pais et al., considers an effect of three different (group, neighbor and single) mice housing conditions during adolescence on the development social and cognitive deficits and anxiety-like behavior in adulthood. The results are interesting and the paper is well written. However, there are some concerns related to the paper.

1.     Methods: A better characterization of differences between each housing condition is needed: i.e., did mice have visual or physical contact, could they smell or hear other mice. 

2.     Results: Some findings indicate that adolescent social isolation might induce a deficit in social contact (Liu J et al., 2015).  It should be discussed in the context of lack of impairments in social interaction in single-housed mice.

3.     Discussion: In my opinion a sentence: … “neighbor housing returned object recognition memory to the levels observed in group housed mice” ( line 343/344) in not correct. Neighbor housing just did not affect recognition memory.

Author Response

Methods: A better characterization of differences between each housing condition is needed: i.e., did mice have visual or physical contact, could they smell or hear other mice. 

We have added text to the Methods better describing the neighbor housing complexes. The mice have limited physical contact due to the wire mesh (with 1cm2 openings), but have complete visual, olfactory and auditory stimulus from the neighboring mice.

Results: Some findings indicate that adolescent social isolation might induce a deficit in social contact (Liu J et al., 2015).  It should be discussed in the context of lack of impairments in social interaction in single-housed mice.

Thank you for this citation. We had missed this reference in our literature survey.  We have added this reference to our Discussion and have included a rationale for why we may have not found a similar effect in the present experiments.

Discussion: In my opinion a sentence: … “neighbor housing returned object recognition memory to the levels observed in group housed mice” ( line 343/344) in not correct. Neighbor housing just did not affect recognition memory.

We have changed sentences within the Discussion to be more specific in our interpretation of the novel object recognition data. We have instead stated that “Neighbor housing or group housing did not alter recognition memory in the novel object task, but single housing lead to recognition deficits.”

Reviewer 2 Report

In this study, Pais and colleagues have examined social interaction and cognitive deficits in socially isolated and group housed mice, as well as (and more importantly) mice housed in a novel neighbor-housing environment. Neighbor-housing increased social interaction, and rescued deficits in novel object recognition. Furthermore, anxiety-like behavior, both social and basal, was reduced in neighbor-housed mice compared to their socially isolated counterparts. The study is well-designed and the novel housing condition is excellent. I only have one minor comment. Total distance travelled is sometimes attributed to increased anxiety-like behavior. Please comment on that.

Author Response

We appreciate you raising this issue. We acknowledge that hyperactivity in an anxiety task can be confounding for the interpretation of the light-dark assay data and have included these statements in the Discussion.